# A prospective study of mental wellbeing, quality of life, human-animal attachment, and grief among foster caregivers at animal shelters

Lauren Powell *, Roxy Ackerman, Chelsea L. Reinhard, James Serpell, Brittany Watson

School of Veterinary Medicine, University of Pennsylvania, Philadelphia, PA, United States of America

* lrpowell@vet.upenn.edu

## Abstract

Foster care programs at animal shelters have emerged as an important tool for promoting animal welfare and supporting shelter life-saving efforts. Preliminary evidence suggests that foster caregiving may also be health-promoting for humans. The goals of this study were to investigate the experiences of foster caregivers at animal shelters based on measures of positive and negative affect, quality of life, and grief, and to describe human-animal attachment among foster populations. Between March 2022 and 2023, 131 foster caregivers from five shelters in the United States completed three online surveys before, during and after providing foster care to a shelter animal. Positive affect decreased significantly from baseline to post-foster (F = 5.71, p<0.01), particularly among dog caregivers (F = 6.17, p<0.01). Negative affect remained unchanged (F = 0.47, p = 0.63). Foster caregivers perceived their foster animal provided companionship, affection and emotional support, although dog foster caregivers reported significantly higher emotional (U = 313.50, p<0.01) and social/physical quality of life (t = 4.42, p<0.01) than cat foster caregivers. Caregivers reported low mean avoidant and anxious attachment, suggesting they were able to develop secure bonds with their foster animals. Retention of fosters was also strong, with 86% of caregivers reporting they were likely to provide foster care in the future. Our findings suggest that fostering at animal shelters may serve as a One Health intervention to offer companionship, affection and emotional support to human caregivers while promoting animal welfare. However, these benefits did not translate to improvements in caregiver mental wellbeing, so caution should be applied when considering foster caregiving as a potential mental health promotion tool.

## Introduction

Millions of animals enter animal shelters around the world each year where they face a host of stressors and, in some cases, a heightened risk of euthanasia [1–5]. Shelters often have limited space and resources, so animals may have a reduced ability to exercise, exhibit normal

**Data Availability Statement:** The dataset supporting this paper is available at Dryad: https://doi.org/10.5061/dryad.5mkkwh7d6.

**Funding:** The study was funded by Nestlé Purina PetCare Global Resources, Inc. The Arnall Family Foundation provided salary support for L.P. and the Bernice Barbour Foundation provided salary support for C.L.R. The funders had no role in the study design, data collection and analysis, decision to publish, or preparation of the manuscript.

**Competing interests:** The authors have declared that no competing interests exist.

behaviors and interact with conspecifics or humans. The loss of social attachments, excess noise, and lack of control or predictability in the shelter can also exaggerate stress levels and reduce quality of life (QoL) (see Taylor and Mills [6] for full review). Foster care has emerged as a promising tool to improve animal welfare for shelter animals. Short-term stays in a foster home can decrease stress, reduce cortisol and promote rest among shelter dogs [7]. Placement in a foster home can also boost animals' visibility in the community, resulting in greater opportunities for adoption [8] and an increased likelihood of leaving the shelter alive [9].

Companion animal foster caregiving may also be health-promoting for humans [10]. A recent, cross-sectional survey of about 600 shelter foster caregivers found almost all caregivers agreed with the statement 'fostering dogs adds to my happiness' and 88% agreed that 'interacting with my fosters helps me stay healthy' [11]. In a pilot study of 11 foster caregivers, after six weeks of caring for a shelter dog, foster caregivers performed more physical activity and spent less time sitting based on accelerometer data. Caregivers also reported fewer depressive symptoms, and half the sample had met new people in their neighborhood because of their foster dog [12]. A second pilot study of four war veterans reported similarly promising results. Dog foster caregivers performed more steps per day, reported better QoL, increased positive affect, lower stress and decreased depression after three months of fostering [13].

However, foster caregivers are likely to also encounter challenges while providing care and/ or feelings of loss when the animal departs the home. Fostering animals with behavioral problems appears to be particularly challenging [11,14], with previous data indicating that 23% of foster caregivers found caring for behavioral cases very emotionally challenging compared with only 7% of medical cases [11]. Caregivers who frequently foster puppies or 'special needs' cases have also reported more thoughts of quitting due to burnout [11]. Research focused on the experiences of human foster parents shows grief around the departure of a child leads many caregivers to consider quitting, although agency support and caregiver training can mitigate the risk of caregiver turnover [15]. Grief among companion animal foster caregivers has received little scientific attention, although initial data suggests that organizational support from shelters may be similarly important in reducing the emotional impacts of fostering and increasing the retention of caregivers [11].

Human-animal bonds could also influence caregivers' perceptions of the benefits and challenges of shelter foster caregiving. Thielke and Udell [16] hypothesized that foster-animal attachment could be a source of stress for foster caregivers who feel conflicted about bonding with their foster animals due to concerns about how they will adjust when separated. The researchers used the LAPS questionnaire [17] to measure the strength of attachment among foster caregivers, finding comparable levels of attachment between animal shelter volunteers and foster caregivers [16]. Using a truncated version of the same questionnaire, Reese, Jacobs [11] also reported foster caregivers had similar levels of attachment to pet owners from previously published studies [17]. In their sample, increased attachment was correlated with a lower risk of quitting fostering due to bad experiences, suggesting that human-animal attachment may alleviate some of the stressors associated with foster caregiving [11].

No studies to date have considered foster caregiver-animal attachment based on style rather than strength, which limits our understanding of the human-animal bond in shelter populations [18,19]. Anxious and avoidant attachment dimensions have been used to characterize attachment in other populations. Individuals who score low on both components are said to have secure attachments where they feel comfortable bonding with and relying on their animal [20]. Anxious attachment describes intrusive thoughts, worrying, and seeking proximity or reassurance from the animal, and has been correlated with increased animal-directed caregiving behaviors [21] but poorer mental wellbeing, and increased anxiety and distress [20]. Avoidant attachment, which describes individuals who maintain emotional distance and avoid intimacy with their pet, has

been associated with less attentiveness towards animals [21] but does not impact mental wellbeing or distress [20]. Service dog foster carers have been shown to exhibit increased avoidant attachment compared with companion animal owners [22,23]. The researchers hypothesized that avoidant attachment may help to prepare caregivers for the eventual departure of the animal [22]. Human attachment styles also impact animal behavior [24–26]. Increasing avoidant attachment, for example, has been associated with a greater prevalence of separation-related distress among dogs [25] which may be particularly relevant for shelter dogs as some evidence suggests they may be at greater risk of exhibiting separation anxiety [27,28].

Public interest in foster caregiving spiked during the COVID-19 pandemic, with reports of increased applications during the lock-down period [29,30]. Many shelters attempted to recruit new foster caregivers during this time and some initiated foster care programs for the first time [31]. However, the increased interest has since subsided [29] so retention of existing caregivers has become even more crucial for shelters. The goal of this study was to characterize the experiences of foster caregivers before, during, and after providing foster care for a shelter animal, using measures of positive and negative affect, QoL, and grief, and to describe retention among shelter foster populations. A secondary aim was to measure human-animal attachment among foster caregivers based on anxious and avoidant attachment dimensions.

## Materials and methods

### Study population

Foster caregivers were recruited from five shelters in the United States between 8th March 2022 and 20th March 2023, including Providence Animal Center in PA, Humane Animal Partners in DE, Dakin Humane Society and Massachusetts Society for the Prevention of Cruelty to Animals in MA, and San Diego Humane Society in CA. The first four shelters were small to medium in size, taking in between 2655 and 5906 animals in 2019, while San Diego Humane Society was much larger, taking in approximately 29,000 animals in 2019 [described further in 32]. To be eligible to participate, foster caregivers had to be over the age of 18 years, agree to provide temporary care to a shelter animal in their home, not have fostered an animal in the previous seven days, not planning to use the foster period as a trial adoption and not have participated in the study previously. The inclusion and exclusion of participants is described further in Fig 1. The study was reviewed and approved by the University of Pennsylvania Institutional Review Board (protocol number 850635) and all participants provided written informed consent.

### Protocol

The study used a prospective, observational design where foster caregivers were invited to participate in the study by shelter staff when they booked an appointment or attended the shelter to pick-up a foster animal (Fig 2). The baseline survey was completed online using Qualtrics up to seven days before the foster animal was taken home or on an iPad at the shelter, depending on the shelter's processes. Researchers then recorded the date on which the animal entered foster care based on the shelter records, which was accessed directly using PetPoint (PetPoint Data Management System, Version 6, Pethealth Software Solutions Inc., USA), Shelterluv (Shelterluv, Inc., USA) or through exported reports provided by the shelters. We contacted the participants via email or phone (depending on the participant's preference) 10 days after the foster animal entered their home to complete the second survey. Researchers tracked when foster animals had an outcome in the shelter records or exported reports and contacted caregivers to complete the third survey once the foster animal had left the caregiver's home. Foster caregivers were sent up to two reminders on subsequent days if they had not completed the follow-up surveys.

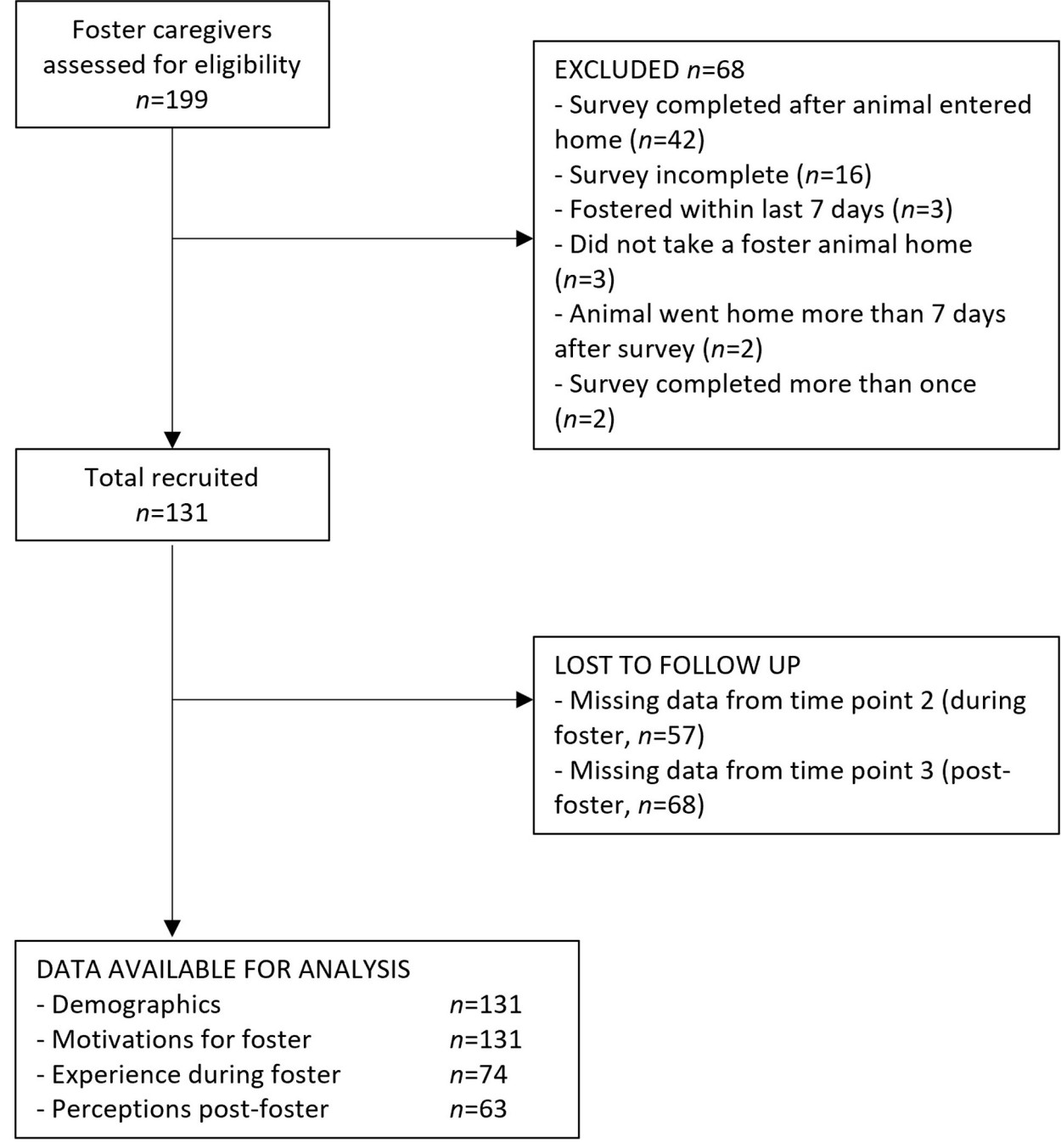

**Fig 1. Flow diagram showing the number of individuals with valid data at each stage.**

## Questionnaires

The results from the baseline survey describing foster caregivers' motivations for care based on their demographic characteristics and personality traits have been described in detail previously [32]. The data from the questionnaires below are presented here for the first time.

**Animal demographics.** Foster caregivers were asked to report their foster animal's shelter ID number, foster species, and primary and secondary reasons for foster at time point 2

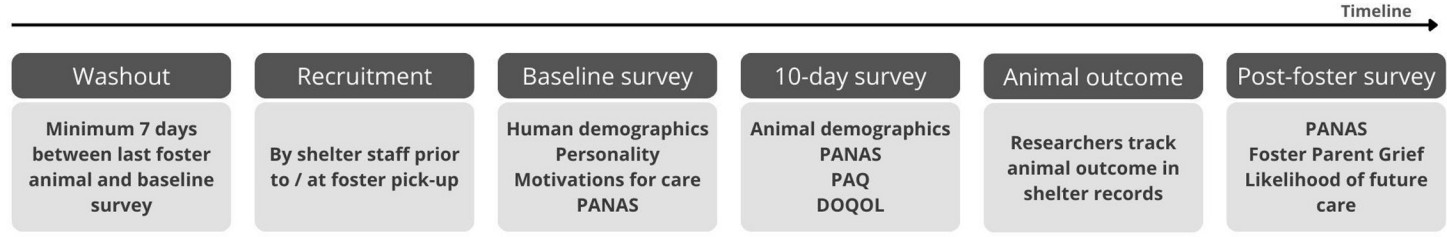

**Fig 2. Study protocol, including questionnaires used at each time point.** PANAS: Positive and Negative Affect Schedule. PAQ: Pet Attachment Questionnaire. DOQOL: Dog Owner-specific Quality of Life questionnaire.

(during care). We also asked participants whether their foster animal was experiencing any health conditions on a four-point scale (no condition, minor, moderate or serious health conditions) and whether they were experiencing any problems with their foster animal's behavior (no problems, minor, moderate or serious problems).

**Positive and Negative Affect Schedule (PANAS).** Participants completed the short form of the Positive and Negative Affect Schedule (PANAS) at each time point to provide measures of affect across the foster experience. The PANAS is a 10-item questionnaire that assesses an individual's recent feelings and mood on a five-point scale from "never" (1) to "always" (5). It produces two subscales based on positive and negative affect, by averaging the respondents' scores for the descriptors 'alert', 'inspired', 'determined', 'attentive', 'active' and 'upset', 'hostile', 'ashamed', 'nervous' and 'afraid', respectively. The short form of the PANAS exhibits strong psychometric properties, including internal reliability, convergent and criterion validity and has been shown to reflect measures of subjective well-being and happiness [33]. The PANAS has also been shown to reflect other measures of mental health. Scores in the positive affect scale are typically negatively correlated while negative affect scores are positively correlated with measures of psychological distress, stress, anxiety, depression, and fatigue [34–36].

**Dog Owner-specific Quality of Life questionnaire (DOQOL).** The Dog Owner-specific Quality of Life questionnaire (DOQOL) was used to characterize the role of the foster animal in foster caregivers' QoL during foster care (time point 2). The survey includes 10 questions presented on a seven-point scale, with answers ranging from "disagree strongly" (1) to "agree strongly" (7). The scores were then averaged to produce three subscale scores for emotional QoL, social/physical QoL, and stress/interference QoL. The DOQOL has established reliability and validity among dog owners but has not yet been applied among cat caregivers [37].

**Pet Attachment Questionnaire (PAQ).** The Pet Attachment Questionnaire (PAQ) is a validated 26-item questionnaire that was used to measure attachment orientations of foster caregivers towards their foster animals during care (time point 2). Individuals were asked to indicate how much they agreed with each prompt on a 7-point Likert scale. Subscale scores for anxious and avoidant attachment were then calculated by reverse scoring the relevant prompts and averaging the scores. A higher score in both subscales was indicative of a higher avoidant or anxious attachment [20].

**Foster parent grief scale.** At time point 3, after the foster animal had an outcome (e.g., adoption, returned to shelter, euthanized), caregivers completed a shortened version of the Foster Parent Grief Scale, adjusted to reflect shelter animals rather than human children. The original scale included a series of 16 statements about anticipatory and experienced grief where foster parents were instructed to indicate how much they agreed with each statement on a 5-point scale from strongly agree (1) to strongly disagree (5) [38]. Here, the foster caregivers were asked 7 adapted questions from the scale and several additional questions about how challenging they found foster care and how likely they were to provide foster care to future shelter animals.

## Statistical analysis

As the DOQOL had not previously been used among cat caregivers, we first ran a confirmatory factor analysis in IBM SPSS AMOS 28 Graphics. Univariate normality was assessed using visual investigation of histograms, skew and kurtosis values and the Shapiro-Wilk test, and multivariate normality was assessed using Mardia's normalized estimate of multivariate kurtosis. The data showed some non-normality, although skew and kurtosis values fell within normal ranges (-2 to +2, and <7 respectively). To account for this, we ran a maximum likelihood (ML) model with bootstrapping including 500 samples [39]. All other statistical analyses were conducted in IBM SPSS Statistics, version 29. We first compared demographic characteristics between participants with valid data across all time points relative to those with missing data using Pearson Chi Square or Fisher-Freeman tests where >20% of cells had an expected frequency less than 5. Linear mixed-effects models were then used to examine changes in positive and negative affect with time point (baseline, 10-day, and post-foster), foster species, an interaction term between time point*foster species, age, gender and previous foster experience included as fixed effects and participant included as a random effect. These models are commonly used for repeated measures as they can accommodate non-independence of measurements between time points and missing data points, so participants with missing data do not need to be excluded on a pairwise basis, thus maximizing sample size and reducing possible bias [40]. The positive affect residuals were normally distributed based on visual inspection of histograms and Q-Q plots. The negative affect residuals showed moderate deviation from normality, although previous research shows linear mixed-effects models are remarkably robust to violations of model assumptions [41]. We then compared QoL and pet attachment relative to foster species, perceived behavioral problems as a binary variable (no problems/minor, moderate or serious problems), the presence of medical conditions (no condition/minor, moderate or serious condition), and previous foster experience using Mann Whitney U tests for non-parametric data and independent samples t-tests for parametric data. Pearson's r and Hedges' g were calculated as measures of effect size, respectively, and are reported alongside p values we previously suggested [42]. P<0.05 was considered statistically significant, unless otherwise indicated.

## Results

### Descriptive characteristics

The final sample included 131 respondents with valid baseline data, 74 with valid data at time point two (during foster care) and 63 with valid data at time point three (post-foster care, Fig 1). Almost two-thirds of the sample were under 39 years of age (42.0% were 18–29 years and 21.4% were 30–39 years). Most foster caregivers were female (n = 117, 89.3%), white (n = 108, 82.4%), previous (n = 120, 91.6%) or current pet owners (n = 77, 58.8%), and just over half had previous foster experience (n = 70, 53.4%). About half the sample fostered dogs (n = 65, 49.6%), while 46.6% fostered cats (n = 61), and 3.8% fostered other species (n = 5). Caregivers with missing data for at least one time point did not differ in gender (X2(3, n = 131) = 2.60, p = 0.53) or race (X2(6, n = 131) = 5.95, p = 0.40) from those with complete data, but were significantly more likely to be 18–29 years old (X2(6, n = 131) = 18.97, p<0.01) and to have no previous experience fostering (X2(1, n = 131) = 5.55, p = 0.02).

Most foster caregivers reported they were not experiencing any problems with their foster animal's behavior (n = 51, 68.9%), although 28.4% experienced minor problems (n = 21), and 2.7% experienced moderate problems (n = 2). No foster caregivers said they were experiencing severe problems with their foster animal's behavior. Health conditions were more common. About half of the fosters reported their animal/s had no health conditions (52.7%, n = 39), but

37.8% reported minor conditions (n = 28), 6.8% had moderate conditions (n = 5) and 2.7% had serious health conditions (n = 2).

## Foster caregiver affect

Linear mixed models revealed a statistically significant main effect of time on positive affect ($F_{2,140.99}$ = 5.71, p<0.01, Table 1) and an interaction effect between time and foster species ($F_{2,140.99}$ = 6.17, p<0.01), suggesting the changes in positive affect differed significantly between dog and cat foster caregivers (Fig 3). Negative affect was comparable at each time point ($F_{2,169.33}$ = 0.47, p = 0.63, Table 1). Age (p≥0.10), gender (p≥0.69) and previous foster experience (p≥0.43) were not significantly associated with positive or negative affect and were subsequently removed from the models. Foster species was also non-significant and thus removed from the negative affect model (p = 0.69).

## Foster caregiver quality of life

The median item scores for the DOQOL are provided in Table 2. Confirmatory factor analysis supported a three-factor model in our sample of dog and cat foster caregivers. The model ($X2$ = 52.31, df = 32, p = 0.01) showed acceptable fit based on the comparative fit index (CFI = 0.95), the Tucker-Lewis Index (TIL = 0.92) and the root mean square error of approximation (RMSEA = 0.09), with all factor loadings in excess of 0.60 (S1 Table). The subscales also retained strong internal reliability based on Cronbach's Alpha (emotional QoL = 0.91, social/physical QoL = 0.83, and stress/interference QoL = 0.81).

A Mann-Whitney U test revealed a small (r = 0.40), statistically significant difference in emotional QoL between dog and cat foster caregivers (U = 313.50, z = -3.39, p<0.01), with dog foster caregivers reporting higher emotional QoL (median = 6.67, IQR = 5.50–7.00) compared with cat foster caregivers (median = 5.33, IQR = 4.33–6.00). We also found a large difference (g = 1.10) in social/physical QoL relative to foster species, with dog foster caregivers reporting a mean score of 4.90 (SD = 1.47) compared with cat caregivers (mean = 3.41, SD = 1.26) based on an independent samples t-test (t = 4.42, p<0.01). Stress/interference QoL did not vary between foster species (t = 0.25, p = 0.80, g = 0.06). Foster caregivers who reported experiencing problems with their foster animal's behavior showed small but non-significant increases in social/physical (t = -0.86, p = 0.39, g = 0.22) and stress/interference QoL (t = -0.83, p = 0.41, g = 0.21) compared with foster caregivers who did not, although emotional QoL was comparable (U = 668.00 z = 0.96, p = 0.34, r = 0.11). Emotional (U = 696.00 z = 0.15, p = 0.88, r = 0.02) and social/physical QoL (t = 0.81, p = 0.42, g = 0.19) were also similar between caregivers who cared for animals with health conditions and those who did not. Stress QoL showed a small, but non-significant (t = 1.78,

**Table 1. Linear mixed models showing effects of time on positive and negative affect scores in response to foster caregiving (n = 131).**

| Variable | Estimated marginal mean | 95% CI | Coefficient | Std error | t |
|---|---|---|---|---|---|
| *Positive affect* | | | | | |
| Baseline | 19.59 | 18.95–20.24 | -0.02 | 0.48 | -0.04 |
| During foster | 19.22 | 18.45–19.99 | -0.23 | 0.51 | -0.46 |
| Post-foster | 18.39 | 17.59–19.19 | Ref | Ref | Ref |
| *Negative affect* | | | | | |
| Baseline | 8.22 | 7.70–8.75 | 0.30 | 0.36 | 0.82 |
| During foster | 7.96 | 7.31–8.62 | 0.03 | 0.40 | 0.09 |
| Post-foster | 7.93 | 7.23–8.63 | Ref | Ref | Ref |

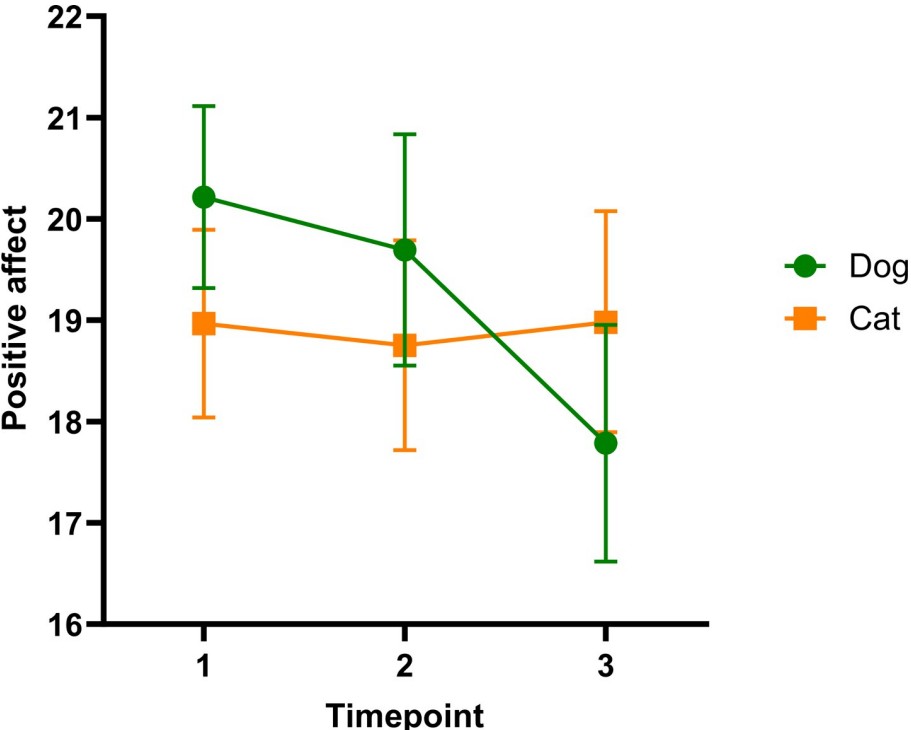

**Fig 3. Estimated marginal means ± 95% confidence intervals showing changes in positive affect before (time point 1), during (time point 2) and after foster care (time point 3) separated by species.**

$p = 0.08$, $g = 0.41$), difference between those facing animal health conditions (mean = 3.51, SD = 1.29) compared with those who were not (mean = 2.93, SD = 1.52). First-time and repeat foster caregivers also reported comparable QoL ($p \geq 0.19$).

**Table 2. Median responses to the DOQOL scale completed during foster care (time point 2, n = 74).**

|  | Median | IQR |
|---|---|---|
| *Emotional QoL (mean = 5.61, SD = 1.28)* |  |  |
| Fostering an animal provides me love and affection | 6.00 | 5.00–7.00 |
| Fostering an animal provides me companionship when I want it | 6.00 | 4.75–7.00 |
| Fostering an animal provides me emotional support | 5.00 | 4.00–7.00 |
| *Social/Physical QoL (mean = 4.02, SD = 1.53)* |  |  |
| Fostering an animal improves the amount of social activities I perform | 4.00 | 3.00–5.00 |
| Fostering an animal improves my ability to do things for fun outside my home | 4.00 | 2.00–5.00 |
| Fostering an animal improves my level of physical activity | 5.00 | 3.00–6.00 |
| *Stress/Interference QoL (mean = 3.21, SD = 1.43)* |  |  |
| Fostering an animal interferes with my other household responsibilities | 3.00 | 2.00–5.00 |
| Fostering an animal results in damage to my belongings or property | 2.00 | 1.00–5.00 |
| Fostering an animal interferes with my ability to go on vacation or leave my house | 4.50 | 1.75–5.00 |
| Fostering an animal increases my level of stress | 2.00 | 1.00–5.00 |

Data shown as median and interquartile range (25th-75th percentile). Possible scores ranged from 1 (strongly disagree) to 7 (strongly agree).

## Foster caregiver attachment to pets

Foster caregivers reported low mean avoidant attachment (mean = 1.90, SD = 0.43, range = 1.08–3.15) and anxious attachment scores (mean = 1.64, SD = 0.48, range = 1.00–3.92) from a possible range of 0 to 7. There were no significant differences based on previous foster experience (p≥0.74), foster species (p≥0.07), caregivers' perceptions of behavioral problems (p≥0.56) or the presence of health conditions (p≥0.07). Although the scores were not statistically different (t = -1.82, p = 0.07), cat foster caregivers reported higher mean avoidant attachment (mean = 1.98, SD = 0.42) than dog foster caregivers (mean = 1.79, SD = 0.43), equivalent to a medium effect size (g = 0.44). Caring for animal/s with a health condition was also associated with a small (r = 0.21), but non-significant, increase in anxious attachment (U = 849.00, Z = 1.81, p = 0.07).

## Grief after foster animal left home

Excluding caregivers who adopted at least one of their foster animals (n = 13), the majority of the sample agreed that they missed their foster animal (42.0% somewhat agreed, 36.0% strongly agreed) and often wondered if the animal was doing well (40.0% somewhat agreed, 52.0% strongly agreed). However, most caregivers also reported that they had adjusted well since their foster animal left (68.0% strongly agree, 22.0% somewhat agree), and a minority felt lonely (18.0% somewhat agree, 0.0% strongly agree) or had periods of tearfulness since their foster animal left (20.0% somewhat agree, 10.0% strongly agree). The experiences of foster caregivers are described further in Fig 4.

## Likelihood of providing future care

Among caregivers with post-foster responses (n = 63), 13 caregivers adopted at least one of their foster animals, 26 had an animal with an adoption outcome, 3 reported at least one

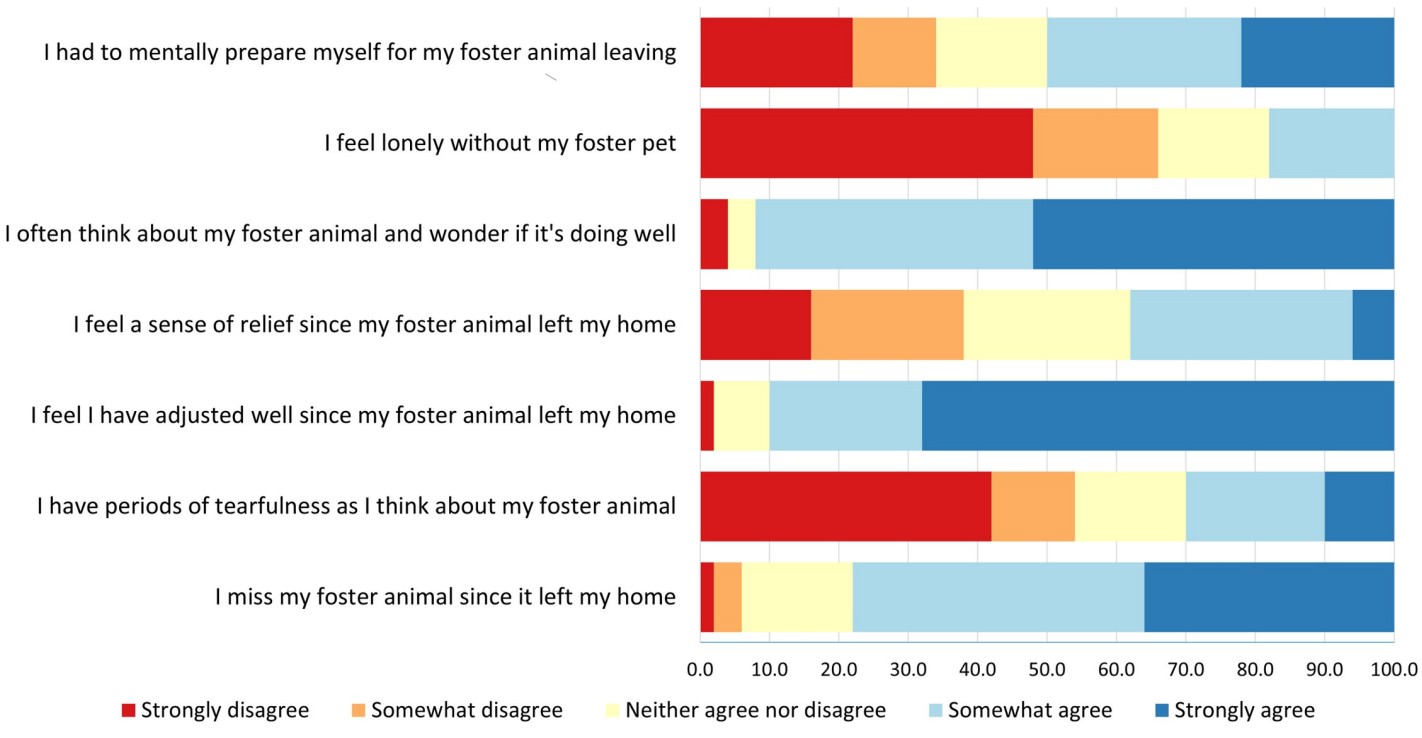

**Fig 4. Foster caregivers' experience after their foster animal left their home (n = 50).**

animal was transferred to another caregiver, 36 returned at least one animal to the shelter, and 2 caregivers reported at least one of their foster animals was euthanized. The majority of foster caregivers were very likely to provide foster care again in the future (68.3%, n = 43), 17.5% were somewhat likely (n = 11), 4.8% were neither likely nor unlikely (n = 3), 6.3% were somewhat unlikely (n = 4) and 3.2% were very unlikely (n = 2). Of the 6 caregivers who were somewhat or very unlikely to provide future care, 50% had adopted their foster animal. Given the small number of participants who were unlikely to provide foster care in the future, we were unable to investigate possible differences in willingness based on previous foster experience, foster species, caregiver perceived behavioral problems, the presence of medical conditions or how challenging the experience was.

## Discussion

Several pilot studies [12,13,43] and a recent narrative review [10] have highlighted foster caregiving at animal shelters as a potential One Health intervention to promote both human and companion animal health and welfare. However, to our knowledge, this is the largest, longitudinal study of foster caregivers' experiences at animal shelters to date.

We found foster caregiving did not promote mental wellbeing, based on repeated measures of positive and negative affect. Negative affect was unchanged throughout the foster period, although caregivers consistently reported fewer negative moods compared with previous studies of shelter staff [44], pet owners [45] and non-clinical populations [33]. Positive affect, on the other hand, decreased significantly after the foster animal left the home relative to baseline levels, particularly among dog foster caregivers. Cat caregivers reported stable levels of positive affect across all time points. The absence of the pet and associated feelings of grief, reported by many participants, likely contribute to the reduction in positive moods following the animal's departure. The varying impacts on positive mood between species may be due to differences in both the degree and nature of human-animal interactions. For example, dog owners typically exhibit more animal-directed communication behaviors, such as petting, hugging, kissing and talking to, and perceive greater social support from their pets than cat owners [46]. These behavioral differences may facilitate the development of human-dog attachments among foster populations and contribute to the decline in positive affect after the dog leaves the foster home.

The lack of significant difference from baseline to during foster could be due to the timing of data collection. The baseline PANAS survey was completed after the foster caregivers had scheduled a foster appointment but before the animal entered their home, so the upcoming foster experience may have already boosted participants' positive moods at the time of the baseline survey. Future studies may consider including an additional baseline data collection before caregivers book a foster appointment to test this hypothesis. Although caring for a foster animal did not have any additional mental health benefits, positive affect levels were comparable to previously reported mean values at all time points throughout the study [33,44,45]. Our findings differ from an existing pilot study that suggested foster caregiving increased positive affect among four veterans [13] which may be attributable to population differences between veterans and shelter foster populations, or differences between individuals who foster to adopt (as seen in the prior study) relative to those fostering without the intention to adopt.

Although we did not find longitudinal evidence of mental health benefits associated with caregiving, fosters reported self-perceived benefits in QoL. Caregivers perceived their foster animal had a positive impact on their emotional QoL during foster by providing love and affection, companionship, and, to a slightly lesser extent, emotional support. Social and physical QoL was mostly unaffected, while caregivers disagreed that fostering increased their stress-related QoL on average, which mirrors previous research [13,43]. Like some previous studies

of pet owners [47–49], we found dog fosters perceived greater emotional, social and physical QoL benefits while caring for the dog than cat foster caregivers which may also contribute to the species-specific decline in positive affect described above. The differences between dog and cat caregivers are again likely due to variations in the nature of human-animal attachment between species. For example, dogs can motivate their caregivers to walk which can boost emotional and physical QoL, and promote social connections within neighborhoods. These species differences may also reflect an artifact of the scale which was originally designed for use among dog owners [37]. QoL did not differ statistically based on caregiver perceptions of behavioral problems or the presence of medical conditions, despite previous evidence to the contrary [50–52]. Separation-related behaviors, for example, have been associated with increased stress QoL among dog owners, while fearfulness has been linked with decreased physical QoL [50]. The temporary nature of foster caregiving and the training and support provided by shelters may have reduced the impacts of behavioral problems on caregiver QoL compared with typical pet ownership. Further research including larger, more diverse human and animal samples is needed to support this hypothesis.

We were also interested in the attachment styles of foster caregivers as increased anxious and avoidant attachment can negatively impact both human mental wellbeing and animal behavior [20,25,53]. Foster caregivers reported low levels of anxious and avoidant attachment compared with previous samples of pet owners [20] and assistance dog puppy raisers [22] which suggests caregivers were able to develop secure attachment bonds with their foster animals. By providing a secure base, the foster caregivers may also help their foster animals, who are known to exhibit more insecure attachment styles than pet dogs [16], to also develop secure bonds. The benefits of this could transcend the foster home to the future adoptive home by increasing the likelihood and speed of bond formation, as described previously among human foster children [54]. Previous studies have also found cat owners exhibited higher levels of avoidant attachment than dog owners, which Zilcha-Mano, Mikulincer and Shaver [20] hypothesized may be attributed to their species-specific behaviors, since cats are often perceived as independent while dogs are believed to be more attentive to their owners. We did not find statistically significant species differences here but this hypothesis warrants further investigation in foster care populations considering the observed difference in mental health and QoL between dog and cat caregivers.

Grief following the departure of the foster child has been associated with thoughts of discontinuing fostering among human foster caregivers [15]. Here, we found half of our sample reported some signs of anticipatory grief, meaning they anticipated experiencing difficulties when the foster animal left the home [38]. Most of the sample also showed ambiguous grief, which describes grief in situations with ambiguity surrounding the loss, such as when an individual is alive but not physically present [55]. For foster carers, this typically manifests in a desire to know how the animal or child is doing [56,57], as we also saw here. However, the majority of caregivers in our study also reported they had adjusted well since their foster animal left the home, which was considerably higher than previous studies of human foster children [38]. Although not a primary aim of the current study, future research could investigate the impact of grief on measures of positive and negative affect relative to foster caregiving using a validated, composite measure of pet grief.

Retention of foster caregivers in this sample appeared promising, with 86% indicating they were likely to provide foster care in the future. In fact, 'foster failing' where caregivers adopted their foster animal appeared to be a common reason why caregivers were unlikely to foster in the future. Foster failing has also been described as a common reason for quitting foster care in previous studies of shelter dogs, typically because participants felt they no longer had room to care for an additional animal [14]. Although we measured future intention to foster rather than

actual behavior, and research shows that people do not always do what they intend to do [58], the findings suggest that most caregivers at least plan to foster shelter animals in the future.

The findings should be considered in light of several limitations. First, we were unable to calculate a response rate as shelter staff informed foster caregivers about the study, so the findings may be subject to selection bias. It is also possible that the results are impacted by nonresponse bias as a considerable proportion of caregivers dropped out of the study. For example, caregivers may have been more motivated to complete the study if they had strong opinions about their foster care experience, including both positive and negative perspectives. The generalizability of the findings is limited as most foster programs were located in the Northeastern United States and the study population was mostly white and female. However, it is also possible that the homogeneity of the study sample is indicative of a broader lack of diversity within shelter foster populations, as has also been described among shelter volunteers [59]. We were also limited by the length of the survey, and thus included positive and negative affect as sole measures of participants' moods. Although affect is correlated with other measures of mental health and wellbeing [34–36], future research may consider using surveys specifically designed to measure stress, anxiety, depression or loneliness to further elucidate the impacts of foster caregiving on human mental health. The inclusion of physiological measures of stress and mental wellbeing, such as cortisol, oxytocin or heart rate variability, could also bolster our study findings.

## Conclusion

We did not find longitudinal evidence to support the notion that foster caregiving can promote mental wellbeing, although both dog and cat foster caregivers reported that their foster animal provided love, affection, companionship and emotional support. Dog caregivers also indicated that their physical and social QoL benefited from foster caregiving. Caregivers appeared to form secure bonds with their foster pets, which could benefit both human mental wellbeing and animal behavior and welfare. While caregivers showed some signs of anticipatory and ambiguous grief at the end of the foster period, most believed they had adjusted well since their animal left and the vast majority were likely to provide foster care to future shelter animals. Our results provide support for the continued expansion of foster care programs for the mutual benefits of caregivers and shelter animals. However, additional studies including more comprehensive measures of mental wellbeing are needed to determine whether companion animal foster caregiving may serve as a mental health promotion tool.

## Supporting information

**S1 Table. Standardized factor loadings of DOQOL from confirmatory factor analysis using a maximum likelihood model with bootstrapping including 500 samples.** (DOCX)

## Acknowledgments

We thank Dakin Humane Society, Providence Animal Center, Humane Animal Partners, Massachusetts Society for the Prevention of Cruelty to Animals, and San Diego Humane Society for their support of the study and their help recruiting foster caregivers.

## Author Contributions

**Conceptualization:** Lauren Powell, Chelsea L. Reinhard, James Serpell, Brittany Watson.

**Data curation:** Lauren Powell, Roxy Ackerman.

**Formal analysis:** Lauren Powell.

**Funding acquisition:** Lauren Powell, Chelsea L. Reinhard, James Serpell, Brittany Watson.

**Methodology:** Lauren Powell.

**Project administration:** Lauren Powell, Roxy Ackerman.

**Resources:** Lauren Powell.

**Software:** Brittany Watson.

**Supervision:** Lauren Powell.

**Visualization:** Lauren Powell.

**Writing – original draft:** Lauren Powell.

**Writing – review & editing:** Lauren Powell, Roxy Ackerman, Chelsea L. Reinhard, James Serpell, Brittany Watson.

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
