## [Decision Letter · Decision Letter 0]

3 Jan 2024

PONE-D-23-34226A prospective study of mental wellbeing, quality of life, the human-animal bond, and grief among foster caregivers at animal sheltersPLOS ONE

Dear Dr. Powell,

Thank you for submitting your manuscript to PLOS ONE. After careful consideration, we feel that it has merit but does not fully meet PLOS ONE’s publication criteria as it currently stands. Therefore, we invite you to submit a revised version of the manuscript that addresses the points raised during the review process. Your submission has been reviewed and I invite you to consider the reviewer comments, which you will find below. I'm asking you to pay special attention to the reviewer's first comment, about which responses are included in the analysis. For the various suggestions for including additional factors in the analysis, please consider them.

We look forward to receiving your revised manuscript.

Kind regards,

I Anna S Olsson, Ph.D.

Academic Editor

PLOS ONE

“The study was funded by Nestlé Purina PetCare Global Resources, Inc. The Arnall Family Foundation provided salary support for L.P. and the Bernice Barbour Foundation provided salary support for C.L.R.”

4. In the online submission form, you indicated that [Data cannot be shared publicly because of our data sharing agreements with the animal shelters. Data are available from the principal author on request.].

Reviewers' comments:

Reviewer's Responses to Questions

**Comments to the Author**

1. Is the manuscript technically sound, and do the data support the conclusions?

Reviewer #1: Partly

2. Has the statistical analysis been performed appropriately and rigorously? 

Reviewer #1: No

3. Have the authors made all data underlying the findings in their manuscript fully available?

Reviewer #1: No

4. Is the manuscript presented in an intelligible fashion and written in standard English?

Reviewer #1: Yes

5. Review Comments to the Author

Reviewer #1: This manuscript describes a study exploring the impact of fostering animals on caregivers' mental wellbeing and quality of life. The introduction provides useful background information, and I find that publishing the “negative result” that fostering animals does not significantly improve caregivers' mental wellbeing – as measured by positive affect – is worthwhile, as it is an important consideration for those interested in fostering based on that assumption. The contrast with the self-perceived increase in mental wellbeing is also an interesting result.

There are, however, a few issues to be addressed before this study can be published. These include the following:

• The study examines positive and negative affect, quality of life, and grief, considering human-animal attachment in the sample which is of considerable size (N=131) but subjected to high (>50%) attrition, which can potentially bias the results. This is adequately addressed, but it is not clear whether the authors only analysed and made comparisons based on the 63 participants that completed the study, and this should be clarified. For example, in table 1 a N=121 is reported for data comprising the three time-points, which makes no sense, given only 63 participants finished the study. Same for table 2, in which an n=72 is reported. In any case, any analysis concerning the impact of fostering animals should be pair-wise, and should thus exclude drop-outs. In that regard, demographics on the relevant sample (those who completed the study) should be provided. Moreover, these results are best analysed by a repeated measures test, which would have the benefit of having more statistical power, yet although it is claimed in ln 287 that this is the case, no repeated measures tests are described in the methods section. Also regarding statistical reporting, it is not acceptable to report any non-statistical difference as a difference. Either there is a statistically significant difference, for the predefined alpha (in this case I’m assuming it is .05 before corrections) or there is no difference.

• The authors report that out of the 6 caregivers who reported to be somewhat or very unlikely to provide foster care, 3 of them had adopted their foster animals. I would exclude these from the analysis on this item. One thing is to have an experience as a foster and after returning the animals stating they do not want to do it again. Another thing altogether is to, after having experienced fostering an animal, keeping that animal. This means that the foster had a positive experience, rather than a negative one.

• I could not find the comparison in QoL tests between before, during and after the fostering experience. Where is that reported in the text?

• In the difference in reported QoL between dog and cat fosters, it would be interesting to explore more possible causes, even if speculatively Possible causes that come to mind are the different degrees (and/or kind) of interaction with humans between the two species, or the fact that dogs must be walked and this results in fosters doing more exercise and interact with their environment , which might have an effect on QoL.

• Another thing that is not explored as a cause for reduced positive affect from fostering (but not between the “before” and “during” time-points) is the loss of the animal itself, after the fostering period, as this might take a toll. The high number of fosters experiencing grief may be an indicator, and it would be interesting to use it as a factor in the analysis.

• Another thing that might be useful is to break the data on the PANAS and DOQOL for the difference species fostered, as there may be an effect in this regard.

• Another key issue to address is the lack of raw data, which should be submitted, according to best practice.

6. PLOS authors have the option to publish the peer review history of their article (what does this mean?). If published, this will include your full peer review and any attached files.

Reviewer #1: No

---

## [Author Response · Author response to Decision Letter 0]

4 Mar 2024

Reviewer #1: This manuscript describes a study exploring the impact of fostering animals on caregivers' mental wellbeing and quality of life. The introduction provides useful background information, and I find that publishing the “negative result” that fostering animals does not significantly improve caregivers' mental wellbeing – as measured by positive affect – is worthwhile, as it is an important consideration for those interested in fostering based on that assumption. The contrast with the self-perceived increase in mental wellbeing is also an interesting result.

Thank you for your time and effort reviewing our manuscript. 

There are, however, a few issues to be addressed before this study can be published. These include the following:

1. The study examines positive and negative affect, quality of life, and grief, considering human-animal attachment in the sample which is of considerable size (N=131) but subjected to high (>50%) attrition, which can potentially bias the results. This is adequately addressed, but it is not clear whether the authors only analysed and made comparisons based on the 63 participants that completed the study, and this should be clarified. For example, in table 1 a N=121 is reported for data comprising the three time-points, which makes no sense, given only 63 participants finished the study. Same for table 2, in which an n=72 is reported. In any case, any analysis concerning the impact of fostering animals should be pair-wise, and should thus exclude drop-outs. In that regard, demographics on the relevant sample (those who completed the study) should be provided. Moreover, these results are best analysed by a repeated measures test, which would have the benefit of having more statistical power, yet although it is claimed in ln 287 that this is the case, no repeated measures tests are described in the methods section. Also regarding statistical reporting, it is not acceptable to report any non-statistical difference as a difference. Either there is a statistically significant difference, for the predefined alpha (in this case I’m assuming it is .05 before corrections) or there is no difference.

We use linear mixed models to analyze positive and negative affect. These models are commonly used for repeated measures as they can account for non-independence of measurements between timepoints. They also have the benefit of accommodating missing data points, meaning we do not need to exclude participants who have missing data at one or more time points, thus maximizing the sample size and reducing possible bias from pairwise deletion. We have added this information under ‘Statistical analysis’ in the methods for readers who are not familiar with linear mixed models. We have also added results describing the demographic characteristics between participants with complete datasets and those with missing data (lines 224-227). 

Table 2 reports quality of life data that was collected at timepoint 2 only (during care) as the survey asks about the impacts of the specific foster animal which could not be answered before or after care. We have added “completed during foster care (timepoint 2)” to the table heading to clarify this point. 

Regarding the statistical reporting, we have included both the effect size and p value as recommended (see Sullivan & Feinn, 2012). We have now added this citation to the methods section. We also double checked the results to ensure all nonsignificant differences are clearly described as such. 

2. The authors report that out of the 6 caregivers who reported to be somewhat or very unlikely to provide foster care, 3 of them had adopted their foster animals. I would exclude these from the analysis on this item. One thing is to have an experience as a foster and after returning the animals stating they do not want to do it again. Another thing altogether is to, after having experienced fostering an animal, keeping that animal. This means that the foster had a positive experience, rather than a negative one.

We agree with the reviewer that the experiences of individuals who are unlikely to foster in the future due to adoption may differ from those who had negative experiences. However, the question was not designed as a proxy measure for whether the foster experience was positive or negative. We were interested in learning about retention of foster caregivers and it appears adoption is a key reason affecting one’s willingness to foster in the future which is important for shelters to consider when developing and managing a foster program. We did exclude the participants from the further analysis looking at willingness to foster based on foster characteristics. However, ultimately this is not included in the final paper as the sample size was too small so we removed the results (see lines 304-307). 

3. I could not find the comparison in QoL tests between before, during and after the fostering experience. Where is that reported in the text?

Please see our response to comment 1 above, and Figure 2. We have also added “(time point 2)” to the methods section describing the QoL scale and when it was completed to clarify this point. 

4. In the difference in reported QoL between dog and cat fosters, it would be interesting to explore more possible causes, even if speculatively Possible causes that come to mind are the different degrees (and/or kind) of interaction with humans between the two species, or the fact that dogs must be walked and this results in fosters doing more exercise and interact with their environment , which might have an effect on QoL.

We have now added several sentences to the discussion describing possible differences in the degree and nature of human-animal interactions between species, with a particular focus on the possible role of dog-walking as a means of boosting mood and QoL (lines 326-331 and 353-357). 

5. Another thing that is not explored as a cause for reduced positive affect from fostering (but not between the “before” and “during” time-points) is the loss of the animal itself, after the fostering period, as this might take a toll. The high number of fosters experiencing grief may be an indicator, and it would be interesting to use it as a factor in the analysis.

We have added an exploration of the possible role of grief in reducing positive affect post-foster to the discussion (lines 324-326). However, we have not included grief as a factor in the analysis as the adapted scale hasn’t been validated for use as a single, composite measure. The scale was included in this study for descriptive purposes only as the length of validated pet grief surveys prevented their inclusion. We have suggested in the discussion (lines 389-391) that future researchers could investigate the impact of grief on measures of positive and negative affect relative to foster caregiving using a validated measure of pet grief in the discussion.

6. Another thing that might be useful is to break the data on the PANAS and DOQOL for the difference species fostered, as there may be an effect in this regard.

The differences in DOQOL based on species are described in the second paragraph of the ‘Foster caregiver quality of life’ section of the results. We have now included foster species and an interaction term between foster species and time point in the PANAS models which revealed differences between dog and cat foster caregivers (see 'Foster caregiver affect' under the results). These results have also been described in the discussion (lines 323-331). 

7. Another key issue to address is the lack of raw data, which should be submitted, according to best practice.

The dataset has now been uploaded to a public data repository. DOI: 10.5061/dryad.5mkkwh7d6. For the purpose of peer review, the data can be accessed at https://datadryad.org/stash/share/foh-FWULRD3e2yxbVE4LDzNZZC1jvNpDW6XYee6CBKc.

---

## [Editor Report · Decision Letter 1]

21 Mar 2024

A prospective study of mental wellbeing, quality of life, human-animal attachment, and grief among foster caregivers at animal shelters

PONE-D-23-34226R1

Dear Dr. Powell,

We’re pleased to inform you that your manuscript has been judged scientifically suitable for publication and will be formally accepted for publication once it meets all outstanding technical requirements.

Kind regards,

I Anna S Olsson, Ph.D.

Academic Editor

PLOS ONE
---

## [Editor Report · Acceptance letter]

25 Apr 2024

PONE-D-23-34226R1 

PLOS ONE

Dear Dr. Powell, 

I'm pleased to inform you that your manuscript has been deemed suitable for publication in PLOS ONE. Congratulations! Your manuscript is now being handed over to our production team.

Kind regards, 

on behalf of

Dr. I Anna S Olsson 

Academic Editor

PLOS ONE